# Systems Analysis Reveals Ageing-Related Perturbations in Retinoids and Sex Hormones in Alzheimer’s and Parkinson’s Diseases

**DOI:** 10.3390/biomedicines9101310

**Published:** 2021-09-24

**Authors:** Simon Lam, Nils Hartmann, Rui Benfeitas, Cheng Zhang, Muhammad Arif, Hasan Turkez, Mathias Uhlén, Christoph Englert, Robert Knight, Adil Mardinoglu

**Affiliations:** 1Faculty of Dentistry, Oral and Craniofacial Sciences, King’s College London, London SE1 9RT, UK; simon.1.lam@kcl.ac.uk; 2Leibniz Institute on Aging-Fritz Lipmann Institute, 07745 Jena, Germany; nils.hartmann@unimedizin-mainz.de (N.H.); christoph.englert@leibniz-fli.de (C.E.); 3National Bioinformatics Infrastructure Sweden (NBIS), Science for Life Laboratory, Department of Biochemistry and Biophysics, Stockholm University, SE-17121 Stockholm, Sweden; rui.benfeitas@scilifelab.se; 4Science for Life Laboratory, KTH—Royal Institute of Technology, SE-17121 Stockholm, Sweden; cheng.zhang@scilifelab.se (C.Z.); muhammad.arif@scilifelab.se (M.A.); mathias.uhlen@scilifelab.se (M.U.); 5Department of Medical Biology, Faculty of Medicine, Atatürk University, 25240 Erzurum, Turkey; hasanturkez@gmail.com; 6Institute of Biochemistry and Biophysics, Freidrich-Schiller-University Jena, 07745 Jena, Germany

**Keywords:** neurodegeneration, Alzheimer’s, Parkinson’s, ageing, systems biology

## Abstract

Neurodegenerative diseases, including Alzheimer’s (AD) and Parkinson’s diseases (PD), are complex heterogeneous diseases with highly variable patient responses to treatment. Due to the growing evidence for ageing-related clinical and pathological commonalities between AD and PD, these diseases have recently been studied in tandem. In this study, we analysed transcriptomic data from AD and PD patients, and stratified these patients into three subclasses with distinct gene expression and metabolic profiles. Through integrating transcriptomic data with a genome-scale metabolic model and validating our findings by network exploration and co-analysis using a zebrafish ageing model, we identified retinoids as a key ageing-related feature in all subclasses of AD and PD. We also demonstrated that the dysregulation of androgen metabolism by three different independent mechanisms is a source of heterogeneity in AD and PD. Taken together, our work highlights the need for stratification of AD/PD patients and development of personalised and precision medicine approaches based on the detailed characterisation of these subclasses.

## 1. Introduction

Neurodegenerative diseases, including Alzheimer’s (AD) and Parkinson’s diseases (PD), cause years of a healthy life to be lost. Much previous AD and PD research has focused on the causative neurotoxicity agents, namely, amyloid β and α-synuclein, respectively. The current front-line therapies for AD and PD are cholinesterase inhibition and dopamine repletion, respectively, which are considered gold standards. Unfortunately, these therapies are not capable of reversing neurodegeneration [1,2], thus necessitating potentially lifelong dependence on the drug and risking drug-associated complications. Moreover, AD and PD are complex multifactorial diseases with heterogeneous underlying molecular mechanisms involved in their progression [3,4,5]. This variability can explain the differences in patient response to other treatments such as oestrogen replacement therapy [6,7] and statin treatment [8,9]. Hence, we observed that there are distinct disease classes affecting specific cellular processes. Therefore, there is a need for the development of personalised treatment regimens.

In this study, we propose a holistic view of the mechanisms underlying the development of AD and PD rather than focusing on amyloid β and α-synuclein [10]. To date, complex diseases including liver disorders and certain cancers have been well studied through the use of metabolic modelling. This enabled the integration of multiple omics data for stratification of patients, discovery of diagnostic markers, identification of drug targets, and proposing of personalised or class-specific treatment strategies [11,12,13,14]. A similar approach may be applied for AD and PD since there is already a wealth of data from AD and PD patients from post-mortem brain tissues and blood transcriptomics.

AD and PD share multiple clinical and pathological similarities, including comorbidities [15,16], inverse associations with cancer [17,18], and ageing as a risk factor [19,20]. One type of ageing is telomeric ageing, which is associated with the loss of telomeres, protein/nucleic acid structures that protect chromosome ends from degradation [21]. The enzyme telomerase is necessary for the maintenance of telomeres. In adults, telomerase activity is mostly limited to progenitor tissues such as in the ovaries, testes, and bone marrow. Loss of telomerase activity leads to telomere shortening, loss of sequences due to end-replication, and eventual degradation of sequences within coding regions, leading to telomeric ageing. Considering AD and PD as products of ageing, we can use an ageing model organism to study its effects on the brain. In our study, we used zebrafish (*Danio rerio*) as a model organism since it has been used extensively used to study vertebrate ageing [22]. For example, a zebrafish ageing model can harbour a nonsense mutation in the *tert* gene, which encodes the catalytic subunit of telomerase, and exhibit faster-than-normal ageing [23,24].

In our study, we first analysed post-mortem brain gene expression data and protein–protein interaction data from the Genotype-Tissue Expression (GTEx) database [25], Functional Annotation of the Mammalian Genome 5 (FANTOM5) database [26,27,28,29], Human Reference Protein Interactome (HuRI) database [30], and Human Protein Atlas (HPA) (http://www.proteinatlas.org, accessed on 9 March 2021) [31] for characterization of normal brain tissue (Figure 1A). Secondly, we analysed transcriptomic data from the Religious Orders Study and Rush Memory Aging Project (ROSMAP) [32,33,34] with published expression data from anterior cingulate cortices and dorsolateral prefrontal cortices of PD and Lewy body dementia patients, hereafter referred to as the Rajkumar dataset [35], and from putamina, substantiae nigrae, and prefrontal cortices from patients with PD, hereafter referred to as the Zhang/Zheng dataset [36,37]. On these data, we conducted differential gene expression and functional analysis, and then constructed biological networks to further explore coordinated patterns of gene expression. Next, we performed global metabolic analyses using genome-scale metabolic modelling. Alongside these analyses, we also leveraged zebrafish *tert* mutants to test the hypothesis that the identified changes may be associated with telomeric ageing. Finally, on the basis of our integrative systems analysis, we defined three distinct disease subclasses within AD and PD and identified retinoids as a common feature of all three subclasses, being likely to be perturbed through ageing. We revealed subclass-specific perturbations at three separate processes in the androgen biosynthesis and metabolism pathway, namely, oestradiol metabolism, cholesterol biosynthesis, and testosterone metabolism.

## 2. Materials and Methods

### 2.1. Data Acquisition and Processing

Gene expression values of protein-coding genes from the ROSMAP dataset were determined using kallisto (version 0.46.1, Pachter Lab, Berkeley, CA, USA) [38] by aligning raw RNA sequencing reads to the *Homo sapiens* genome in Ensembl release 96 [39]. Raw single-cell RNA sequencing reads from ROSMAP were converted to counts in Cell Ranger (version 4.0, 10x Genomics, Pleasanton, CA, USA, https://support.10xgenomics.com/single-cell-gene-expression/software/pipelines/latest/installation; accessed on 24 July 2020), and aligned to the Cell Ranger *Homo sapiens* reference transcriptome version 2020-A. Single-cell expression values were compiled into pseudo-bulk expression profiles for each sample.

Expression values of protein-coding genes from brain samples of the ROSMAP dataset [32,33,34], GTEx database version 8 [25], FANTOM5 database [26,27,28] via Regulatory Circuits Network Compendium 1.0 [29], HPA database [31], Rajkumar dataset [35], and Zhang/Zheng dataset [36,37] were then combined. Genes from GTEx and FANTOM5 brain samples were filtered such that only genes whose products are known to participate in a protein–protein interaction described in the HuRI database [30] were included. Expression values were scaled and TMM normalised per sample, Pareto scaled per gene, and batch effects removed with the *removeBatchEffect* function from the limma (version 3.42.0, The Walter and Eliza Hall Institute of Medical Research, Parkville, Australia) [40] R package. After quality control and normalisation, a total of 64,794 genes and 2055 samples resulted. As the data also included samples from patients with neurological conditions other than AD or PD, we then removed those samples and finally accepted 1572 samples corresponding to AD, PD, or control for further analysis.

Projections onto 2-D space by PCA, t-SNE [41], and UMAP [42] methods were generated on data after missing value imputation with data diffusion [43]. t-SNE projections were generated with perplexity 20 and 1000 iterations. All other parameters were kept default. PCA and UMAP projections were generated using all default parameters.

### 2.2. Transcriptome Analysis

Using normalised, imputed expression values, AD and PD samples were then arranged into clusters without supervision using ConsensusClusterPlus (version 1.50.0, University of North Carolina at Chapel Hill, Chapel Hill, NC, USA) [44] with maxK = 20 and rep = 1000. All other parameters were kept default. Clustering by *k* = 3 clusters was selected for downstream analysis. A fourth cluster containing only control samples was artificially added to the analysis.

For differential gene expression analysis, normalised, non-imputed counts were used. Genes were removed if expression values were missing in 40% or more of samples or were zero in all samples. Differential expression was then performed using DESeq2 (version 1.26.0, European Molecular Biology Laboratory, Heidelberg, Germany) [45] with uniform size factors and all other parameters set to default. Genes with a Benjamini–Hochberg adjusted *p*-value at or below a cut-off of 1 × 10^−10^ were determined significantly differentially expressed genes.

Gene set enrichment analysis was performed using piano (version 2.2.0, Chalmers University of Technology, Göteborg, Sweden) [46] using all default parameters. GO term lists were obtained from Ensembl Biomart (https://www.ensembl.org/biomart/martview, accessed on 9 March 2021) and were used as gene set collections. Enrichment of GO terms was determined by analysing GO terms of genes differentially expressed genes detected by DESeq2 as well as the parents of those GO terms. GO terms with an adjusted *p*-value at or below 0.05 for distinct-directional and/or mixed-directional methods were determined statistically significant.

### 2.3. Metabolic Analysis

For each cluster, consensus gene expression values were determined by taking the arithmetic mean of normalised expression counts across all samples within each cluster.

A reference GEM was created by modifying the gene associations of all reactions within the adipocyte-specific GEM *iAdipocytes1850* [47] to match those within the generic human GEM HMR3 [48]. The resulting GEM was designated *iBrain2845*. Cluster-specific GEMs were reconstructed using the RAVEN Toolbox (version 2.0, Chalmers University of Technology, Göteborg, Sweden) [49] tINIT algorithm [50,51], with *iBrain2845* as the reference GEM.

FBA was conducted on each cluster-specific GEM using the *solveLP* function from the RAVEN Toolbox with previously reported constraints [52] and defining ATP synthesis (*iBrain2845:* HMR_6916) as the objective function. All constraints were applied with the exception of the following reaction IDs, which were excluded: EX_ac[e] (*iBrain2845*: HMR_9086) and EX_etoh[e] (*iBrain2845*: HMR_9099).

Reporter metabolite analysis was conducted using the *reporterMetabolites* function [53] from the RAVEN Toolbox, using *iBrain2845* as the reference model.

### 2.4. Network Analysis

To generate gene networks, we took normalised, non-imputed expression values from AD and PD samples. Control samples and samples from blood were excluded. One network was generated each for AD and PD. For the AD network, all male samples were included, and 171 female samples were chosen at random and included. For the PD network, all samples were included. Genes with any missing values were dropped. Genes with the 15% lowest expression or 15% lowest variance were disregarded from further analysis. Spearman correlations were calculated for each pair of genes, and the top 1% of significant correlations were used to generate gene co-expression networks. Random Erdős–Rényi models were created for the AD and PD networks, with the same numbers of nodes and edges to act as null networks, and compared against their respective networks in terms of centrality distributions. Community analyses were performed through the Leiden algorithm [54] by optimising CPMVertexPartition, after a resolution scan of 10,000 points between 10^−3^ and 10. The scan showed global maxima at resolutions = 0.077526 and 0.089074 for AD and PD networks, respectively, which were used for optimisation. Enrichment analysis was performed on modules with >30 nodes using enrichr (https://maayanlab.cloud/Enrichr, accessed on 5 March 2021) [55,56] using GO Biological Process, KEGG, and Online Mendelian Inheritance in Man libraries and was explored using Revigo (http://revigo.irb.hr, accessed on 5 March 2021) [57].

### 2.5. Zebrafish Data Acquisition and Analysis

The *tert* mutant zebrafish line (*tert*^hu3430^) was obtained from Miguel Godhino Ferreira [24]. Fish maintenance, RNA isolation, processing, and sequencing were conducted as described previously [58].

From *n* = 5 wild-type (*tert*^+/+^), *n* = 5 heterozygous mutant (*tert*^+/−^), and *n* = 3 homozygous mutant (*tert*^−/−^), expression values were determined from RNA sequencing reads using kallisto by aligning to the *Danio rerio* genome in Ensembl release 96 [39]. Expression values were generated for each extracted tissue as well as ‘psuedo–whole animal’, containing combined values across all tissues.

A reference zebrafish GEM was manually curated by modifying the existing *ZebraGEM2* model and was designated *ZebraGEM2.1*.

Differential expression analysis, gene set enrichment analysis, GEM reconstruction, FBA, and reporter metabolite analysis were conducted on *tert*^−/−^ and *tert*^+/−^ animals against a *tert*^+/+^ reference using DESeq2, piano, and RAVEN Toolbox 2.0 with default parameters. Reporter metabolite analysis was conducted with *ZebraGEM2.1* as the reference GEM.

FBA was attempted as described for the human GEMs with the exception that the following metabolic constraints were excluded: r1391, HMR_0482 (*ZebraGEM2.1*: G3PDm), EX_ile_L[e] (*ZebraGEM2.1*: EX_ile_e), EX_val_L[e] (*ZebraGEM2.1*: EX_val_e), EX_lys_L[e] (*ZebraGEM2.1*: EX_lys_e), EX_phe_L[e] (*ZebraGEM2.1*: EX_phe_e), GLCt1r, EX_thr_L[e] (*ZebraGEM2.1*: EX_thr_e), EX_met_L[e] (*ZebraGEM2.1*: EX_met__L_e), EX_arg_L[e] (*ZebraGEM2.1*: EX_arg_e), EX_his_L[e] (*ZebraGEM2.1*: EX_his__L_e), EX_leu_L[e] (*ZebraGEM2.1*: EX_leu_e), and EX_o2[e] (*ZebraGEM2.1*: EX_o2_e). The objective function was defined as ATP synthesis (*ZebraGEM2.1*: ATPS4m). FBA results for zebrafish are not presented.

### 2.6. Data and Code Accessibility

All original computer code, models, and author-curated data files have been released under a Creative Commons Attribution ShareAlike 4.0 International Licence (https://creativecommons.org/licenses/by-sa/4.0/; accessed on 29 March 2021) and are freely available for download from <https://github.com/SimonLammmm/ad-pd-retinoid>; accessed on 29 March 2021.

Zebrafish *tert* mutant sequencing data have been deposited in the NCBI Gene Expression Omnibus (GEO) and are accessible through GEO Series accession numbers GSE102426, GSE102429, GSE102431, and GSE102434.

### 2.7. Ethics Statement

Zebrafish were housed in the fish facility of the Leibniz Institute on Aging—Fritz Lipmann Institute (FLI) under standard conditions and a 14 h light and 10 h dark cycle. All animal procedures were performed in accordance with the German animal welfare guidelines and approved by the Landesamt für Verbraucherschutz Thüringen (TLV), Germany.

## 3. Results

### 3.1. Stratification of Patients Revealed Three Distinct Disease Classes

We retrieved gene expression and protein–protein interaction data from GTEx, FANTOM5, HuRI, HPA, and ROSMAP databases and integrated these data with the published datasets by Rajkumar and Zhang/Zheng. After performing quality control and normalisation (as outlined in the Materials and Methods), we included a total of 629 AD samples, 54 PD samples, and 889 control samples in the analysis (Table 1). To reveal transcriptomic differences between AD/PD samples compared to healthy controls, we identified differentially expressed genes (DEGs) and performed gene set enrichment (GSE) analyses. However, since AD and PD are complex diseases with no single cure, it is likely that multiple gene expression profiling exist, manifesting in numerous disease classes requiring distinct treatment strategies. We therefore used unsupervised clustering to elucidate these expression profiles and stratify the AD and PD patients on the basis of the underlying molecular mechanisms involved in the disease occurrence.

Following unsupervised clustering with ConsensusClusterPlus [44], we separated AD and PD samples into three clusters (Figure 1B and Appendix A). Clusters 1 and 2 contained samples from Zhang/Zheng and Rajkumar datasets, respectively, in addition to samples in the ROSMAP dataset, and consisted of 127 and 186 samples from female donors, respectively, and 73 and 95 samples from male donors, respectively. Cluster 2 also contained 14 samples with sex not recorded. Cluster 3 contained only ROSMAP samples and consisted of 114 female and 74 male samples. Clusters did not form firmly along lines of sex, age, or brain tissues or brain subregion (Appendix A). Samples from non-diseased individuals were artificially added as a fourth, control cluster, consisting of 495 female samples, 262 male samples, 13 samples with sex not recorded, and 119 samples derived from aggregate sources.

By differential expression analysis using DESeq2 [45], we then characterised the distinct transcriptomic profiles within our disease clusters (Figure 2A). Cluster 1 showed mixed up- and downregulation of genes compared to control, whereas cluster 2 showed more downregulation and cluster 3 showed vast downregulation of genes compared to control.

To infer the functional differences between the subclasses, we performed GSE analysis using piano [46] (Figure 2B, Appendix A). Globally, DEGs in any cluster 1–3 were enriched in upregulated Gene Ontology (GO) terms for immune response, olfaction, retinoid function, and apoptosis, but downregulated for copper ion transport and telomere organisation, compared to the control cluster. Considering individual clusters, cluster 1 DEGs were enriched in upregulated GO terms associated with immune signalling, cell signalling, and visual perception. We also found downregulation of GO terms associated with olfactory signalling and cytoskeleton. DEGs in cluster 2 were found to be enriched in downregulated GO terms associated with the cytoskeleton, organ development, cell differentiation, retinoid metabolism and response, DNA damage repair, inflammatory response, telomere maintenance, unfolded protein response, and acetylcholine biosynthesis and binding. On the other hand, we did not find any significantly enriched upregulated GO terms. In cluster 3, we found that DEGs were enriched in upregulated GO terms associated with neuron function, olfaction, cell motility, and immune system. DEGs in cluster 3 were found to be enriched in downregulated GO terms associated with DNA damage response, ageing, and retinoid metabolism and response.

The difference in expression profiles illustrate highly heterogeneous transcriptomics in AD and PD and that there are notable commonalities and differences between the subclasses of AD or PD samples. Interestingly, we found retinoid metabolism or function to be a common altered GO term in all subclasses. This was upregulated in cluster 1 but downregulated in clusters 2 and 3. We therefore observed that retinoid dysregulation appears to be a common ageing-related hallmark in AD and PD.

### 3.2. Metabolic Analysis Revealed Retinoids and Sex Hormones as Significantly Dysregulated in AD and PD

On the basis of clustering and GSE analysis, we identified distinct expression profiles, but these alone could not offer insights into metabolic activities of brain in AD and PD. To determine metabolic changes in the clusters compared to controls, we performed constraint-based genome-scale metabolic modelling. We reconstructed a brain-specific genome-scale metabolic model (GEM) based on the well-studied HMR2.0 [47] reference GEM by overlaying transcriptomic data from each cluster and applying brain-specific constraints as described previously [52] using the tINIT algorithm [50,51] within the RAVEN Toolbox 2.0 [49]. We generated a brain-specific GEM (*iBrain2845*) (Appendix A) and used it as the reference GEM for reconstruction of cluster-specific GEMs in turn. We constructed the resulting context-specific *iADPD* series GEMs *iADPD1*, *iADPD2*, *iADPD3*, and *iADPDControl*, corresponding to cluster 1, cluster 2, cluster 3, and the control cluster, respectively (Appendix A).

We conducted flux balance analysis (FBA) by defining maximisation of ATP synthesis as the objective function. *iADPD1* and *iADPD2* both showed upregulation of fluxes in reactions involved in cholesterol biosynthesis and downregulation in O-glycan metabolism, with reaction flux changes being more pronounced in *iADPD2* than in *iADPD1* (Table 2, Appendix A). We found that the fluxes in *iADPD1* were uniquely upregulated in oestrogen metabolism and the Kandustch–Russell pathway. *iADPD2* was uniquely upregulated in cholesterol metabolism, whereas *iADPD3* uniquely displayed roughly equal parts upregulation and downregulation in several pathways, including aminoacyl-tRNA biosynthesis; androgen metabolism; arginine and proline metabolism; cholesterol biosynthesis; galactose metabolism; glycine, serine, and threonine metabolism; and N-glycan metabolism.

In particular, we observed increased positive fluxes through reactions HMR_2055 and HMR_2059 in *iADPD1*, which convert oestrone to 2-hydroxyoestrone and then to 2-methoxyoestrone (Figure 3). In *iADPDControl*, these reactions carried zero flux. In *iADPD2*, we observed increased positive fluxes through HMR_1457 and HMR_1533, which produce geranyl pyrophosphate and lathosterol, respectively. Both of these molecules are precursors to cholesterol, and while we did not see a proportionate increase in the production of other molecules along the pathway (namely, farnesyl pyrophosphate and squalene), we did observe a general increase in fluxes through the androgen biosynthesis and metabolism pathway. Finally, we observed that *iADPD3* displayed a decreased production of testosterone from 4-androstene-3,17-dione via HMR_1974, despite an increase in production of 4-androstene-3,17-dione via HMR_1971.

Taken together, the obtained results indicate the existence of three distinct metabolic dysregulation profiles in AD and PD, with dysregulation being most pronounced in cluster 2 patients and least pronounced in cluster 3 patients. Furthermore, we found that all three clusters show dysregulations in or around sex hormone biosynthesis and metabolism, which might explain the heterogeneity in responses to sex hormone replacement therapy in AD and PD patients as extensively reported previously [6,59,60,61]. We also confirmed that dysregulations through sex hormone pathways in the *iADPD* series GEMs were not due to differences in relative frequencies between sexes in the main clusters 1-3 (Fisher’s exact test, *p* = 0.4700).

In addition to metabolic inference and FBA, we performed reporter metabolite analysis [53] by overlaying DEG analysis results onto the reference GEM to identify hotspots of metabolism (Table 3, Appendix A). In short, we uniquely identified oestrone as a reporter metabolite in cluster 1, and lipids such as acylglycerol and dolichol in cluster 2. No notable reporter metabolites were identified as significantly changed in cluster 3 only. In common to all clusters 1–3, retinoids, and sex hormones such as androsterone and pregnanediol were identified as significantly changed reporter metabolites, which are generally in line with GSE and FBA results.

### 3.3. Network Analysis Supported Retinoid and Androgen Dysregulation and Suggests Transcriptomic Similarity between AD and PD

To further explore the gene expression patterns shown across AD and PD patients, we took expression data and constructed a weighted gene co-expression network for each group (Spearman ρ > 0.9, FDR < 10^−9^; see the Materials and Methods section). Each network was compared against equivalent randomly generated networks acting as null models. After quality control, the AD network contained 4861 nodes (genes) and ≈397,000 edges (significant correlations), and the PD network contained 5857 nodes and ≈394,000 edges (Figure 4A,B, Table 4). A community analysis to identify modules of highly co-expressed genes [54] highlighted 9 and 15 communities with significant functional enrichment in AD and PD, respectively.

In the AD network, gene module C3 was enriched for genes involved with neuron and synapse development, similar to patient cluster 3; C4 for genes involved with mRNA splicing, similar to patient cluster 2; and C5 for genes involved with the mitochondrial electron transport chain (Figure 4C, Appendix A). C1 and C2 were the gene modules with the largest number of genes. C1 was enriched for gene expression quality control genes and development and morphogenesis genes, mirroring patient cluster 2, whereas C2 contained cytoskeleton-related genes, similar to patient cluster 1.

In the PD network, C1 was enriched for genes involved with retinoid metabolism, glucuronidation, and cytokine signalling. Since androgens are major targets of glucuronidation [62], these results are in line with our main findings. Further, C2 contained DNA damage response and gene regulation genes, similar to patient cluster 2; C3 contained nuclear protein regulation genes; and C4 contained mRNA splicing genes, again similar to patient cluster 2.

Further, the two networks share a large number of enriched terms in common, and there is high similarity between the major gene modules, highlighting the similarity between AD and PD. In addition to this, enrichment analysis for KEGG terms was unable to assign “Alzheimer disease” and “Parkinson disease” to the correct gene modules from the respective networks, and additional neurological disease terms such as “Huntington disease” and “amyotrophic lateral sclerosis” were also identified by the analysis, further suggesting the transcriptomic similarity between neurological diseases. We found that AD C1 and PD C2 were frequently annotated with these disease terms, and these gene modules are also highly similar. Therefore, this gene module could constitute a core set of dysregulated genes in neurodegeneration.

Taken together, the network analysis supports our GSE findings. The functional consequences of differential expression in the patient clusters could be explained by differential modulation of gene modules identified in our network analysis together with dysregulation of a core set of genes implicated in both AD and PD.

### 3.4. Zebrafish Transcriptomic and Metabolic Investigations Suggest an Association between Brain Ageing and Retinoid Dysregulation

To further validate our findings regarding the differences between clusters of human AD and PD samples, we analysed transcriptomic data from *tert* mutant zebrafish and reconstructed tissue-specific GEMs (Figure 5A). To ascertain that these effects of ageing were limited to the brain, we analysed the brain, liver, muscle, and skin of zebrafish as well as the whole animal.

We first repeated DEG and GSE analyses in the *tert* mutants using brain transcriptomic data. We found significant enrichment of GO terms associated with retinoid metabolism as well as eye development and light sensing, in which retinoids act as signalling molecules [63] (Figure 5B and Appendix A, Appendix A). To further support our findings, we then reconstructed mutant- and genotype-specific GEMs by overlaying zebrafish *tert* mutant transcriptomic data onto a modified generic *ZebraGEM2* GEM [64]. We designated the modified GEM *ZebraGEM2.1* (Appendix A) and used it as the reference GEM. We also generated zebrafish organ-specific GEMs and provided them to the interested reader (Appendix A).

We then repeated reporter metabolite analysis using the transcriptomic data from zebrafish tissue-specific GEMs and found that retinoids were identified as significant reporter metabolites in *tert*^+/−^ zebrafish (*p* = 0.045) but not in *tert*^−/−^, where evidence was marginal (*p* = 0.084) (Figure 5C, Table 5, Appendix A). We also observed this result in the skin of *tert*^-/-^ mutants, where evidence was significant (*p* = 0.017). This result can be explained due to the susceptibility of skin as an organ to photoageing, for which topical application of retinol is a widely used treatment [65]. However, we did not find evidence for significant changes in pregnanediol, and androsterone was significant only in the skin of *tert*^−/−^ zebrafish (*p* = 0.017). This would suggest that either change in sex hormones are not ageing-related with regards AD and PD, or the changes were outside the scope of the zebrafish model that we used.

Taken together, these results indicated that ageing can largely explain alterations in retinoid metabolism in the brain but not alterations in sex hormone metabolism. These results also suggest that ageing has a differential effect on different organs, implying that metabolic changes due to ageing in the brain are associated with neurological disorders.

## 4. Discussion

In this work, we integrated gene expression data across diverse sources into context-specific GEMs and sought to identify and characterise disease subclasses of AD and PD. We used unsupervised clustering to identify AD/PD subclasses and employed DEG and GSE analysis to functionally characterise them. We used network exploration, constraint-based metabolic modelling, and reporter metabolite analysis to characterise flux and metabolic perturbations within basal metabolic functions and pathways. We then leveraged expression data from zebrafish ageing mutants to validate our findings that these perturbations might be explained by ageing. Our analysis concluded with the identification and characterisation of three AD/PD subclasses, each with distinct functional characteristics and metabolic profiles. All three subclasses showed depletion of retinoids by an ageing-related mechanism as a common characteristic.

We believe that a combined analysis that integrates AD and PD data is necessary to elucidate common attributes between the two diseases. However, we realised that such an analysis will likely obscure AD- and PD-specific factors, such as amyloid β and α-synuclein, but should aid the discovery of any factors in common. Since AD and PD share numerous risk factors and comorbidities such as old age, diabetes, and cancer risk, we believe that an AD/PD combined analysis can identify factors in common to both diseases and prove valuable for the identification of treatment strategies that might be effective in the treatment of both diseases.

GSE analysis highlighted significant changes related to retinoid function or visual system function, in which retinol and retinal act as signalling molecules [63], in all clusters (Figure 2, Appendix A). Together with the identification of multiple retinol derivatives as significant reporter metabolites in *iBrain2845* (Table 3, Appendix A), we hypothesised that retinoids are a commonly dysregulated class of molecules in both AD and PD, and that this may be due to an ageing mechanism. Indeed, in our investigation with zebrafish telomerase mutants, we again found alterations in retinoid and visual system function in GSE analysis (Figure 5B and Appendix A, Appendix A) and reporter metabolite analysis (Figure 5C, Table 5, Appendix A).

Retinoids were identified as a reporter metabolite in all three clusters of patients in this study. Further, our zebrafish analysis highlighted the importance of retinoids in ageing of the brain and the skin (Figure 5C, Table 5, Appendix A). Retinol, its derivatives, and its analogues are already used as topical anti-ageing therapies for aged skin [65], and there is a growing body of evidence suggesting its efficacy for the treatment of AD [66,67,68,69]. We add to the body of evidence with this in silico investigation involving zebrafish telomerase mutants, suggesting that the source of retinoid depletion in AD and PD is ageing-related. Interestingly, regarding our finding for skin ageing in zebrafish, lipid biomarkers have been proposed in a recent skin sebum metabolomics study in PD patients [70]. This could be interpreted as co-ageing in brain and skin tissues, possibly allowing for cheap, non-invasive prognostic testing for PD.

In addition to retinoids, we found evidence for subclass-specific dysregulation within the androgen metabolism pathway in each of the three clusters in FBA (Table 2, Appendix A) and reporter metabolite analysis (Table 3, Appendix A). We found that *iADPD1* displayed increased oestrone conversion to the less potent [71] 2-methoxyoestrone, *iADPD2* displayed increased production of the cholesterol precursor molecules geranyl pyrophosphate and lathosterol and increased androgen biosynthesis, and *iADPD3* displayed decreased conversion of 4-androstene-3,17-dione to testosterone. However, there was no definitive evidence to suggest an ageing-related basis for these observations on the basis of our zebrafish study, but this may be due to the diverse functional roles that sex hormones have, limitations within the *ZebraGEM2.1* model, or absence of an actual biological link between sex hormones and ageing of the brain. Despite this, given the widely reported variability in responses to sex hormone replacement therapy in AD and PD [6,8,60,61], we believe that this observation represents a possible explanation for the heterogeneity. Our observation regarding the dysregulation of the androgen pathway at three separate points suggests that dysregulation at other points might also be linked to AD and PD, thus implying that androgen metabolism dysregulation in general might be important for the development of AD and PD. Our finding via network community analysis of a gene module associated with glucuronidation activity points to a possible therapeutic strategy to combat androgen dysregulation. In our study, the limitation of our dataset that some samples were aggregate samples or did not record the donor’s sex meant that we were unable to assess in detail the sex-dependency of our results. However, as has been extensively studied in the AD model mouse [72,73,74], this remains an important question, and more work is needed to elucidate the importance of sex hormones and glucuronidation regarding AD and PD.

Identification of subclasses is desirable to address the heterogeneity in disease with regards transcriptomic profile and treatment response, but patients must be stratified in order to be diagnosed with the correct disease subclass and therefore administer the appropriate treatment. To this end, we used GSE analysis to functionally characterise the AD/PD subclasses (Figure 2, Appendix A). Cluster 2, which was associated with a decreased immune and stress response, appeared to be most severe disease subclass, whereas cluster 3, which was associated with an increased sensory perception of smell, reduced haemostasis, and reduced immune and DNA damage response, seemed to be the least severe. Meanwhile, cluster 1 was associated with an increased immune and inflammatory responses and reduced sensory perception of smell. The functional terms are supported by community analysis of our AD and PD gene co-expression networks, which identified gene modules that roughly align with the GSE results (Figure 4, Appendix A). The proposed severity ratings are supported by FBA findings, which show *iADPD2* as having the highest total flux dysregulation compared to control, and *iADPD3* as having the least (Table 2, Appendix A). Although we did not attempt to characterise for stratifying and diagnosing patients in our study, our findings clearly show that such stratification is possible.

In this work, we leveraged samples from zebrafish telomerase mutants and insights from the *ZebraGEM2.1* metabolic model. These models were utilised in order to test the hypothesis that the observations we made in the human subjects could be explained by telomeric ageing in a way where we could control the ‘dosage’ of ageing, i.e., *tert^+/−^* and *tert^−/−^* mutants. However, it is important to acknowledge the limitations of such models. Although zebrafish are a widely used model organism to study vertebrate ageing [22], most neurons of the brain do not divide and are therefore not likely to be subject to the direct effects of telomeric ageing. Therefore, we cannot conclude that the dysregulations we observed in AD and PD were caused by ageing of neurons per se, but rather correlations exist between telomeric ageing in zebrafish and AD and PD in humans, and these correlations may act via an indirect mechanism affecting multiple systems of the ageing organism. Due to these limitations, more data and more studies are required to support the link between ageing and neuronal degeneration in AD and PD. The basic requirement of such studies would be brain tissue from donors of a wide range of ages. In our study, we utilised datasets containing aggregated samples, where age cannot be assigned, and samples from age-matched donors, meaning that younger donors were poorly represented in our data, making it unsuitable for an ageing analysis.

In conclusion, we report three distinct subclasses of AD and PD. The first subclass was identified as being associated with increased immune response, inflammatory response, and reduced sensory perception of smell, according to GSE results. We observed that this subclass exhibited increased oestradiol turnover, according to FBA results. The second subclass was linked with increased cholesterol biosynthesis and general increased flux through the androgen biosynthesis and metabolism pathway. This subclass was characterised by reduced immune response. The third subclass was characterised by enrichment of GO terms indicating increased sensory perception of smell, reduced haemostasis, and reduced immune and DNA damage response. This subclass also exhibited reduced testosterone biosynthesis from androstenedione, as determined by FBA. All subclasses exhibited dysregulation within the retinoid metabolism pathway. For all subclasses of AD and PD, more investigation is required to verify the effectiveness of these stratification methods and to aid prediction of effective precision therapies. To our knowledge, this is the first meta-analysis at this scale highlighting the potential significance of retinoids, oestradiol, and testosterone in AD and PD by studying the two diseases in combination. We observed that the existence of disease subclasses demands precision or personalised medicine and explains the heterogeneity in AD and PD response to single-factor treatments.

## Figures and Tables

**Figure 1 biomedicines-09-01310-f001:**
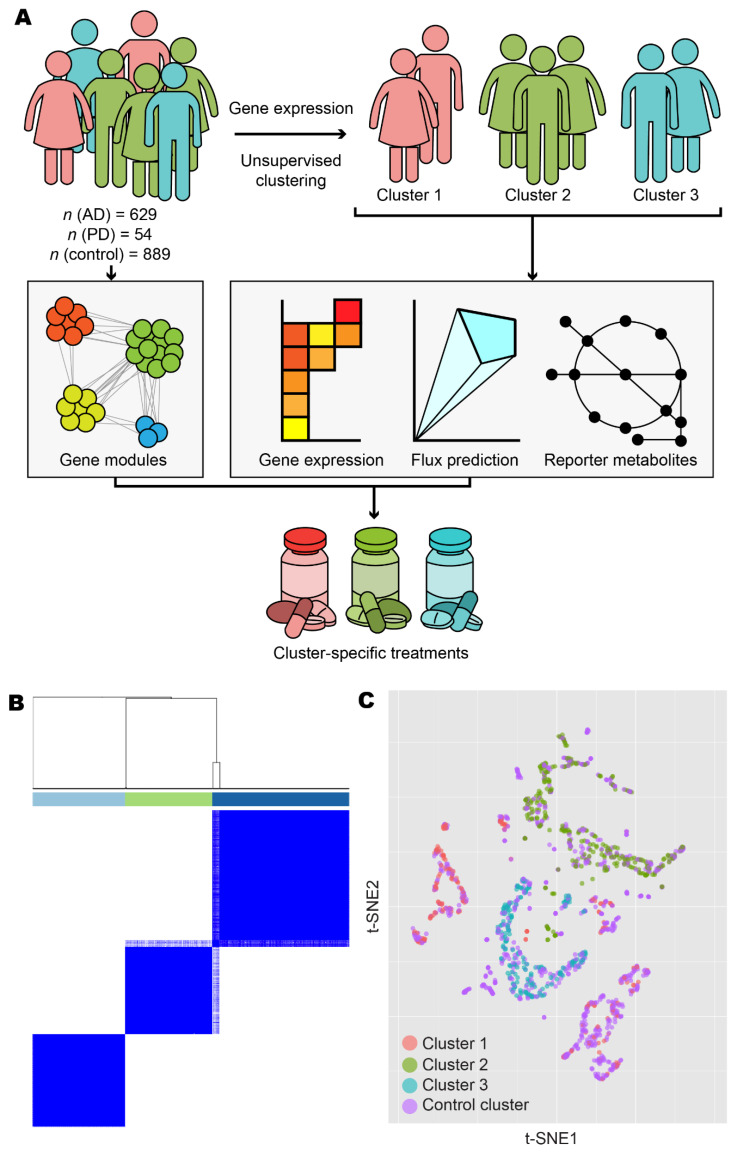
Overview and exploratory data analysis. (**A**) Workflow for the analysis of human AD and PD samples. (**B**) AD and PD samples were clustered into *k* clusters without supervision on the basis of normalised expression counts. Results are shown for *k* = 3 and 1000 bootstrap replicates. Colour bars indicate cluster identity for each sample. For 2 ≤ *k* ≤ 7, refer to Appendix A. (**C**) Normalised expression data from AD, PD, and control samples were projected onto 2-D space using t-distributed stochastic neighbour embedding (t-SNE). Points are coloured according to cluster assignment by unsupervised clustering. For further data visualisation, refer to Appendix A.

**Figure 2 biomedicines-09-01310-f002:**
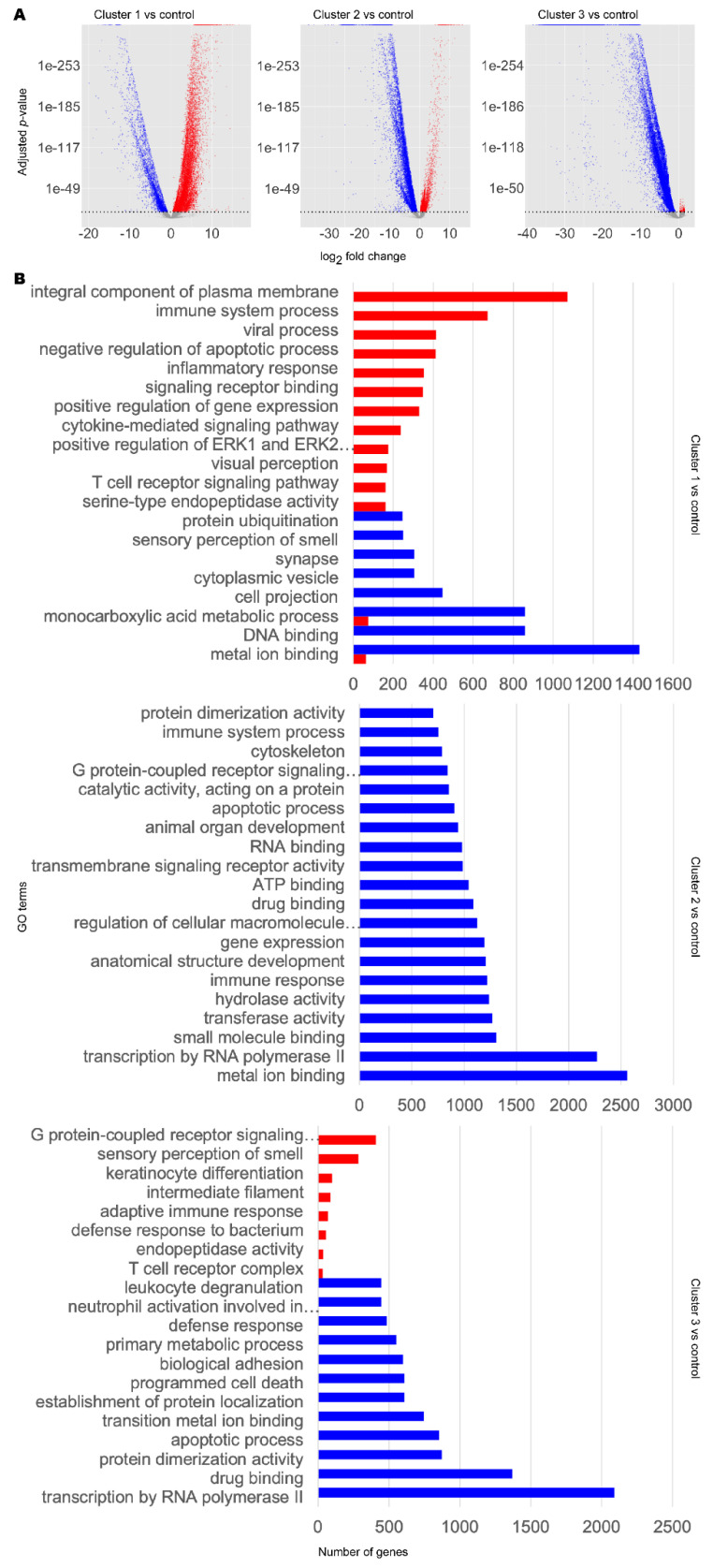
Transcriptomic and functional characterisation of AD and PD subclasses. Differentially expressed gene (DEG) analysis and gene set enrichment (GSE) analysis were performed for AD and PD and control samples for each disease cluster, using the control cluster as reference. (**A**) DEG results. Significant DEGs were determined as those with a Benjamini–Hochberg adjusted *p*-value at or below a cut-off of 1 × 10^−10^. Upregulated significant DEGs are coloured red. Downregulated significant DEGs are coloured blue. Non-significant DEGs are coloured grey. (**B**) Selected significantly enriched GO terms by number of genes as determined by GSE analysis. Red bars indicate upregulated GO terms. Blue bars indicate downregulated GO terms. For full data, refer to Appendix A.

**Figure 3 biomedicines-09-01310-f003:**
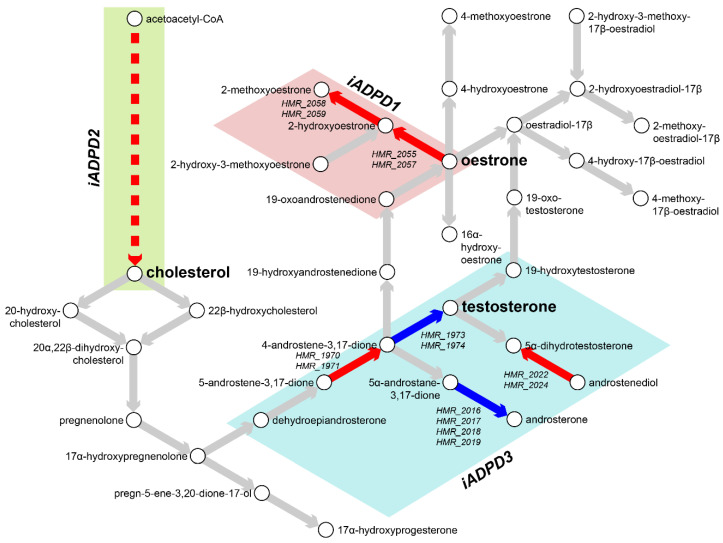
Metabolic characterisation of AD and PD subclasses. Flux balance analysis (FBA) was performed on *iADPD1-3* genome-scale metabolic models (GEMs), and flux values were compared with those of *iADPDControl*. Key metabolites and reactions within the androgen metabolism pathway are shown and key dysregulations are displayed as coloured arrows: red indicates increased flux compared to *iADPDControl*; blue indicates decreased flux compared to *iADPDControl*. Dysregulations associated to each GEM are shown in coloured boxes. The dashed line indicates multiple reactions are involved. Human Metabolic Reactions (HMR) identifiers are shown for androgen metabolism reactions with dysregulated fluxes. For full data, refer to Appendix A.

**Figure 4 biomedicines-09-01310-f004:**
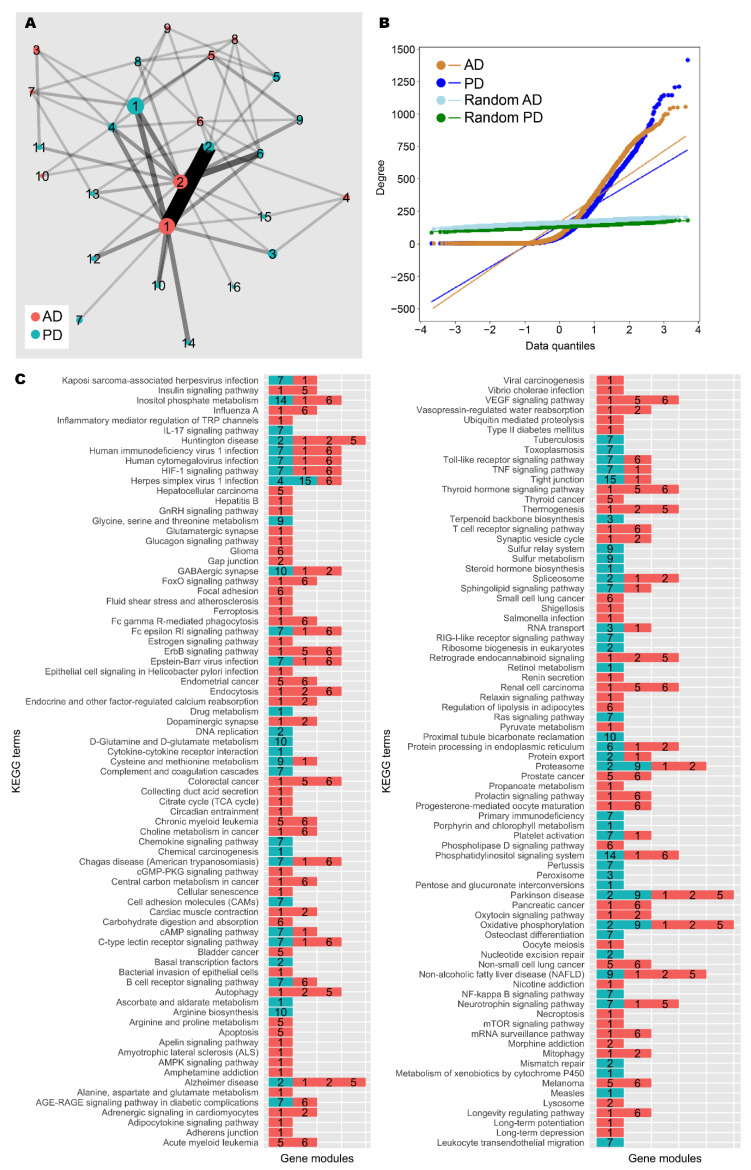
Network analysis of AD and PD gene co-expression modules. (**A**) Gene co-expression networks were constructed from transcriptomic data from AD and PD samples. Community analysis was used to identify gene modules (see the Materials and Methods section). Modules with at least 30 genes are shown as nodes. Node size indicates number of genes. Nodes are coloured by network of origin and numbered in descending order of module size. Shared genes between modules are shown as edges. Edge weight indicates number of shared genes. (**B**) Degree distribution of AD, PD, and random networks. (**C**) Enrichment analysis was performed on gene modules containing at least 30 genes using the KEGG database (see the Materials and Methods section). Significantly enriched gene modules are shown as coloured, numbered blocks. Colour and number keys are as in (**A**).

**Figure 5 biomedicines-09-01310-f005:**
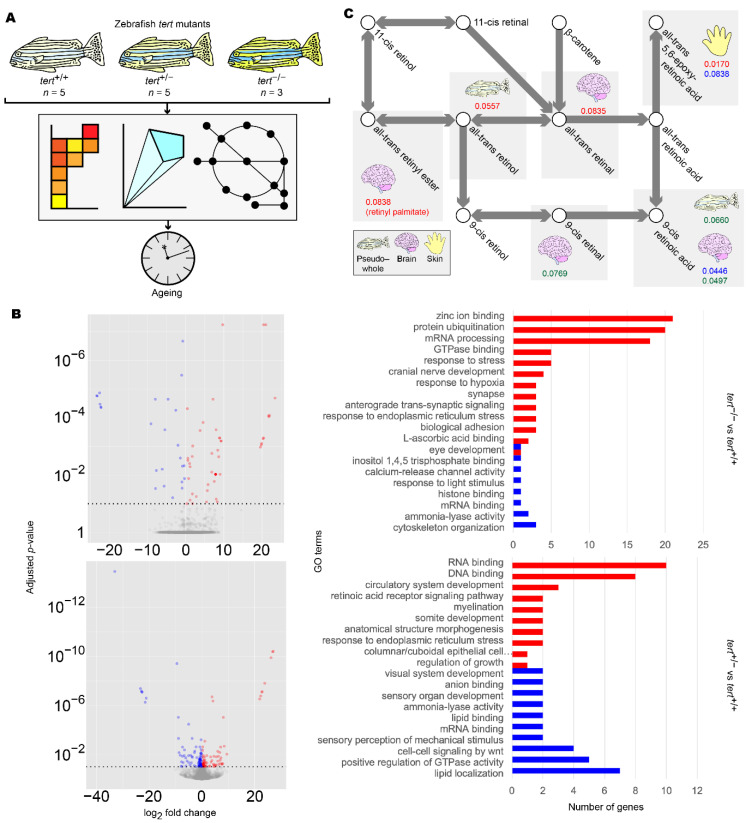
Summary of zebrafish *tert* mutant analysis. (**A**) Workflow for the analysis of zebrafish *tert* mutants. (**B**) Differentially expressed gene (DEG) (left panels) and gene set enrichment (GSE) analysis (right panels) of zebrafish brain samples. DEG and GSE analyses were performed on zebrafish *tert* mutant brain expression data for *tert*^−/−^ (upper panels) and *tert*^+/−^ (lower panels), using *tert*^+/+^ as a reference. Methods and colour keys are as in Figure 2. For muscle, liver, skin, and pseudo-whole animal analyses, refer to Appendix A. For full data, refer to Appendix A. (**C**) Reporter metabolite analysis of zebrafish samples. DEG data were overlaid on *ZebraGEM2.1* to determine reporter metabolites. Shown are reporter metabolites with *p* < 0.1 within the retinoic acid metabolic pathway. Red numbers indicate *p*-values in *tert*^−/−^ compared to *tert*^+/+^. Blue numbers indicate *p*-values in *tert*^+/−^ compared to *tert*^+/+^. Green numbers indicate *p*-values in *tert*^−/−^ compared to *tert*^+/−^. Tissues are indicated with icons. For full data, refer to Appendix A.

**Table 1 biomedicines-09-01310-t001:** Summary of expression data sources.

Source	AD Samples	PD Samples	Control Samples
GTEx/FANTOM5	0	0	67
HPA	0	0	52
Rajkumar	0	14	13
ROSMAP	629	0	704
Zhang/Zheng	0	40	53
Total	629	54	889

Expression data from AD and PD samples were obtained from the Genotype-Tissue Expression (GTEx) database, Functional Annotation of the Mammalian Genome 5 (FANTOM5) database, Human Protein Atlas (HPA), Religious Orders Study and Rush Memory Aging Project (ROSMAP), Rajkumar dataset, and Zhang/Zheng dataset.

**Table 2 biomedicines-09-01310-t002:** Flux balance analysis of *iADPD1*, *iADPD2*, and *iADPD3* versus *iADPDControl*.

Subsystem	*iADPD1*	*iADPD2*	*iADPD3*
Acyl-CoA hydrolysis	−0.001	0.001	0.000
Alanine, aspartate, and glutamate metabolism	−0.148	0.014	0.000
Aminoacyl-tRNA biosynthesis	4.698	4.698	0.000
Androgen metabolism	−1.426	−0.399	−0.001
Arachidonic acid metabolism	−0.098	0.010	0.000
Arginine and proline metabolism	−0.182	−0.327	0.000
Beta oxidation of branched-chain fatty acids (mitochondrial)	−0.049	−0.049	−0.049
Beta oxidation of di-unsaturated fatty acids (n-6) (mitochondrial)	−0.636	0.002	−0.001
Beta oxidation of odd-chain fatty acids (mitochondrial)	0.001	−0.002	−0.002
Beta oxidation of poly-unsaturated fatty acids (mitochondrial)	0.709	0.024	0.000
Beta oxidation of unsaturated fatty acids (n-7) (mitochondrial)	−0.016	0.001	−0.003
Beta oxidation of unsaturated fatty acids (n-9) (mitochondrial)	0.011	0.000	0.007
Carnitine shuttle (cytosolic)	0.012	0.000	−0.001
Carnitine shuttle (mitochondrial)	0.003	0.000	0.002
Cholesterol biosynthesis 1 (Bloch pathway)	0.076	−0.983	0.001
Cholesterol biosynthesis 2	2.501	4.472	0.000
Cholesterol biosynthesis 3 (Kandustch–Russell pathway)	1.699	0.000	0.000
Cholesterol metabolism	0.067	4.482	0.000
Estrogen metabolism	2.085	0.000	0.000
Fatty acid activation (endoplasmic reticular)	0.000	0.000	0.000
Fatty acid biosynthesis (even-chain)	0.000	0.000	0.000
Fatty acid desaturation (even-chain)	0.785	0.000	0.000
Fatty acid elongation (odd-chain)	−0.042	−0.024	0.000
Formation and hydrolysis of cholesterol esters	−0.382	0.004	0.000
Fructose and mannose metabolism	−0.211	−0.007	0.000
Galactose metabolism	−0.008	0.035	0.000
Glycine, serine, and threonine metabolism	0.276	0.557	0.000
Glycolysis/gluconeogenesis	−0.213	0.022	0.033
Histidine metabolism	0.000	0.000	0.000
Leukotriene metabolism	−0.032	0.000	0.000
Lysine metabolism	0.000	0.000	0.000
N-glycan metabolism	−0.784	0.016	0.000
Nitrogen metabolism	0.000	0.000	0.000
Nucleotide metabolism	0.027	−0.028	0.000
O-glycan metabolism	−2.346	−4.738	0.000
Pentose phosphate pathway	0.127	0.000	0.000
Propanoate metabolism	−0.116	0.020	0.091
Protein degradation	0.000	0.000	0.000
Purine metabolism	0.112	−0.013	0.000
Pyrimidine metabolism	−0.071	−0.010	−0.001
Pyruvate metabolism	−0.183	−0.004	−0.077
Starch and sucrose metabolism	0.000	0.000	0.000
Steroid metabolism	−0.097	−0.295	0.003
Terpenoid backbone biosynthesis	0.398	0.187	0.020
Valine, leucine, and isoleucine degradation	0.127	0.000	0.000

Flux balance analysis was performed for each *iADPD*-series GEM, and the predicted fluxes for the three disease cluster GEMs were compared against the predicted fluxes for the control cluster GEM. Reactions are grouped by subsystem and flux difference values are expressed as mean flux difference between disease clusters and the control cluster across all changed reactions within a subsystem. For full results, refer to Appendix A.

**Table 3 biomedicines-09-01310-t003:** Reporter metabolite analysis of AD and PD subclasses.

Reporter Metabolite	Z-Score	*p*-Value
Cluster 1		
O2	6.111	4.95 × 10^−^^10^
Estrone	5.4557	2.44 × 10^−^^8^
Retinoate	5.3943	3.44 × 10^−^^8^
NADP+	5.3667	4.01 × 10^−^^8^
Arachidonate	5.2822	6.38 × 10^−^^8^
2-Hydroxyestradiol-17beta	5.0999	1.70 × 10^−^^7^
Linoleate	5.0622	2.07 × 10^−^^7^
10-HETE	5.0454	2.26 × 10^−^^7^
11,12,15-THETA	5.0454	2.26 × 10^−^^7^
11,14,15-Theta	5.0454	2.26 × 10^−^^7^
Cluster 2		
1-Acylglycerol-3P-LD-PC pool	4.3322	7.38 × 10^−^^6^
Acyl-CoA-LD-PI pool	4.143	1.71 × 10^−^^5^
Phosphatidate-CL pool	4.0973	2.09 × 10^−^^5^
Thymidine	3.5852	0.00016843
Uridine	3.5852	0.00016843
Prostaglandin D2	3.2144	0.00065348
G10596	3.1354	0.0008581
G10597	3.1354	0.0008581
D-Myo-inositol-1,4,5-trisphosphate	2.9988	0.0013552
Dolichyl-phosphate	2.9655	0.001511
Cluster 3		
D-Myo-inositol-1,4,5-trisphosphate	2.6543	0.0039734
13-cis-Retinal	2.6537	0.0039806
Heparan sulfate, precursor 9	2.5915	0.0047772
sn-Glycerol-3-phosphate	2.578	0.0049682
DHAP	2.5353	0.0056173
Porphobilinogen	2.4987	0.0062333
ATP	2.4838	0.0064998
L-Glutamate 5-semialdehyde	2.4576	0.006994
Prostaglandin D2	2.451	0.0071221
ribose	2.4133	0.0079045

Reporter metabolite analysis was performed for each AD/PD subclass by overlaying differential expression results onto *iBrain2845*. Top 10 unique reporter metabolites by *p*-value for each cluster compared to the control cluster are shown. For full results, refer to Appendix A.

**Table 4 biomedicines-09-01310-t004:** AD and PD network properties.

	Nodes	Edges	Diameter	Average Path Length	Density	Clustering Coefficient	Connected Network?	Minimum Cut
AD	4861	396,985	11	3.004	0.034	0.443	No	-
PD	5857	394,405	18	3.598	0.023	0.397	No	-
Random AD	4861	396,985	3	1.970	0.034	0.034	Yes	114
Random PD	5857	394,405	3	2.021	0.023	0.023	Yes	89

Gene co-expression networks were generated for AD and PD samples. AD, PD, and random networks are shown.

**Table 5 biomedicines-09-01310-t005:** Reporter metabolite analysis of zebrafish *tert* mutants.

Reporter Metabolite	Z-Score	*p*-Value
*tert* ^−/−^		
H+	3.911	4.60 × 10^−^^5^
H2O	3.0672	0.0010804
L-Lysine	2.8564	0.0021424
Biocyt c	2.8564	0.0021424
Ubiquinone	2.5742	0.0050241
Nicotinamide adenine dinucleotide—reduced	2.3946	0.0083183
Phosphate	2.0562	0.019883
Superoxide anion	2.0365	0.020851
Sodium	1.9228	0.027254
TRNA (Glu)	1.8752	0.030381
Thiosulfate	1.7684	0.038493
Selenate	1.7684	0.038493
Reduced glutathione	1.7184	0.042862
ADP	1.6716	0.047305
L-Lysine	1.6625	0.04821
Benzo[a]pyrene-4,5-oxide	1.6042	0.054333
Formaldehyde	1.5955	0.055302
L-Glutamate	1.4622	0.071837
(1R,2S)-Naphthalene epoxide	1.4518	0.073276
Aflatoxin B1 exo-8,9-epozide	1.4518	0.073276
*tert* ^+/−^		
H+	4.9585	3.55 × 10^−^^7^
Ubiquinol	3.9938	3.25 × 10^−^^5^
H2O	3.2078	0.00066883
Nicotinamide adenine dinucleotide—reduced	3.029	0.0012268
Superoxide anion	2.0908	0.018274
L-Lactate	2.0752	0.018983
O2	1.9958	0.022976
Lnlncgcoa c	1.9628	0.024834
Succinate	1.9449	0.025895
Ferricytochrome c	1.8352	0.033237
Phosphatidylinositol-3,4,5-trisphosphate	1.7494	0.040109
9-cis-Retinoic acid	1.7	0.044567
[(Gal)2 (GlcNAc)4 (LFuc)1 (Man)3 (Asn)1’]	1.6672	0.047739
O-Phospho-L-serine	1.6601	0.048451
[(Glc)3 (GlcNAc)2 (Man)9 (Asn)1’]	1.6276	0.051802
Protein serine	1.6078	0.053937
[(GlcNAc)1 (Ser/Thr)1’]	1.6078	0.053937
Geranyl diphosphate	1.5912	0.055785
CTP	1.5625	0.059088
[(Gal)2 (GlcNAc)4 (LFuc)1 (Man)3 (Neu5Ac)2 (Asn)1’]	1.5367	0.062179

Reporter metabolite analysis was performed for the brains of zebrafish *tert* mutant by overlaying differential expression results onto *ZebraGEM2.1*. Top 20 unique reporter metabolites by *p*-value for each cluster compared to wild-type *tert*^+/+^ zebrafish are shown. For full results, refer to Appendix A.

## Data Availability

All original computer code, models, and author-curated data files have been released under a Creative Commons Attribution ShareAlike 4.0 International Licence (https://creativecommons.org/licenses/by-sa/4.0/) and are freely available for download from <https://github.com/SimonLammmm/ad-pd-retinoid>. Zebrafish tert mutant sequencing data have been deposited in the NCBI Gene Expression Omnibus (GEO) and are accessible through GEO Series accession numbers GSE102426, GSE102429, GSE102431, and GSE102434.

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
