# Peer review of "Systems Analysis Reveals Ageing-Related Perturbations in Retinoids and Sex Hormones in Alzheimer’s and Parkinson’s Diseases"

_biomedicines, 2021, doi:10.3390/biomedicines9101310_

Round 1

Reviewer 1 Report

Using an in silico database mining approach, this study addresses whether the neurodegenerative diseases Alzheimer's disease and Parkinson's disease display differences in gene expression compared to controls. Furthermore, it is attempted to identify subgroups of patients based on differences in gene expression. By comparison to data from a zebra fish terminal transferase ageing model, the authors try to delineate changes attributed to ageing processes.

The approach of the study and the data retrieval using various software appears to be warranted. Some details could have been described more explicit (example: page 4 line 121-123: After quality control and normalization... it is not clear, how this was done and what led to the final acceptance for analysis). My major concern is the zebra fish ageing model because ageing by terminal transferase deficits applies mainly to dividing cells. After development, most neurons do not divide and ageing of neurons is therefore not well represented by a terminal transferase model. Alzheimer's and Parkinson's disease, however, result from neuronal deficits. Ageing effects could have been better identified by analyzing different age groups within the controls. Differences in disease progression according to gender are well described for Alzheimer's and Parkinson's disease. This study does not differentiate the patient groups according to gender and the composition of the groups according to gender is not described.

The conclusions drawn from the data are sometimes overinterpreted (example page 11 line 285-290: retinoid metabolism upregulation in one cluster and downregulation in two clusters leads to the suggestion that it is a "common ageing-related hallmark of NDD"). Furthermore, observation of expression differences does not indicate whether these are causal, correlated, or counter reaction, however, the authors suggest therapies e.g. page 21 line 493 a retinoid therapy and page 22 line 549 and following several more therapeutic approaches.

In conclusion, this study is based on valid data but needs to consider further analysis:

- expression profiles should be analyzed in different age groups in the control to identify potential ageing related effects

- groups should be differentiated according to gender and separately analyzed (may be important for role of sex hormones)

- suggestions for therapies based on differences in expression profiles alone should be avoided

- the zebra fish data do not contribute much insight to the study

Reviewer 2 Report

Dear authors, greetings for your manuscript. The present manuscript describes retinoids as a key ageing-related feature in AD/PD and identifies three distinct metabolic dysregulation profiles in AD/PD.

Only a very few corrections:

-lines 344-347 "we found that all three feature dysregulations in or associated with sex hormone biosynthesis and metabolism, which might explain the heterogeneity in responses to sex hormone replacement therapy in AD and PD patients as extensively reported previously" there is something wrong in this sentence

-line 464 characterise

-Please, check the manuscript for furthers errors.

Reviewer 3 Report

In this study, a meta-analysis with transcriptomic data of patients suffering either from Alzheimer’s disease or Parkinsons disease in addition to healthy control subjects has been performed along with a co-analysis with an ageing-related zebrafish model. The study is interesting and the manuscript basically carefully written. Please, explain the colour codes for subject groups in Fig. 1A i.e. what are the blue, green and pink subjects. Clarify also whether all subjects in the pink group were males. That is not evident from the Graphical Abstract either, which is otherwise visual with adequate font sizes and restricted amounts of text.

Round 2

Reviewer 1 Report

In the revision, the authors addressed my concerns sufficiently.

I would have appreciated a detailed analysis comparing age and gender related effects. However, the authors address these points now sufficiently in the discussion. Furthermore, the zebra fish model is more carefully annotated and the particular caveats are commented. The suggestions for therapeutical interventions were deleted. Technical details were added.

Author Response

We thank the reviewer for their time checking the revision. We regret being unable to address age- and sex-related effects with the current dataset, but we agree that its discussion is important. We have amended the manuscript to reflect this. We thank the reviewer once more for their valuable time and comments.